# Teenage Mothers in Yaoundé, Cameroon—Risk Factors and Prevalence of Perinatal Depression Symptoms

**DOI:** 10.3390/jcm10184164

**Published:** 2021-09-15

**Authors:** Laure Nicolet, Amir Moayedoddin, Joel Djatché Miafo, Daniel Nzebou, Beat Stoll, Emilien Jeannot

**Affiliations:** 1Institute of Global Health, Faculty of Medicine, Chemin de Mines 9, 1202 Geneva, Switzerland; laure.nicolet@etu.unige.ch (L.N.); Beat.stoll@unige.ch (B.S.); 2Association Actions en Santé Publique, NGO, 1211 Geneva, Switzerland; amir.moayedoddin@amge.ch (A.M.); joel1darchi@gmail.com (J.D.M.); 3Child and Adolescent Psychiatric Service, Faculty of Medicine, Geneva University Hospital (HUG), 1205 Geneva, Switzerland; 4Uni-Psy et Bien-Être (UNIPSY), Yaoundé P.O. Box 35579, Cameroon; danielnzebou@unipsy.cm; 5Addiction Medicine, Department of Psychiatry, Lausanne University Hospital and University of Lausanne, 1004 Lausanne, Switzerland

**Keywords:** teenage mothers, perinatal depression, EPDS score, psychosocial care, Cameroon

## Abstract

Background: Perinatal depression is defined as a non-psychotic depressive episode occurring during pregnancy or during the first year following childbirth. This depressive disorder is highly prevalent among teenage women but there is a lack of data in low- and middle-income countries. The objective of this study was to provide baseline data on the sociodemographic characteristics of pregnant teenagers or teenage mothers in an urban zone in Yaoundé, Cameroon. Risk factors were assessed, and prevalence of depressive disorders was determined. Methods: Women aged 20 years old or less in the perinatal period were invited to participate in the study. A total of 1344 women participated in the four-stage data collection process involving a questionnaire including questions on sociodemographic background, an assessment of their risk of perinatal depression using the EPDS questionnaire (Edinburgh Postnatal Depression Scale), a clinical interview based on the DSM 5 (Diagnostic and Statistical Manual of Mental Disorders), and a final section focusing on risk factors of perinatal depression. Results: The EPDS score was obtained for 1307 women. The prevalence of depressive disorder symptoms among teenage or young pregnant women is estimated to be 70.0%. This risk is significantly increased by different factors including unintended or unplanned pregnancy (aOR: 1.33, 1.14–1.56 CI95%), being separated or single (aOR: 1.34, 1.12–1.60 CI95%), experiencing depression and anxiety before childbirth (aOR: 1.50, 1.02–2.27 CI95%), abortion experience (aOR: 2.60, 1.03–7.14 CI95%) and domestic violence (aOR: 1.76, 1.12–2.83 CI95%). Conclusion: The results of this study reveal a high prevalence of depressive disorder symptoms within the study population. These findings highlight the need to develop maternal care programs to support both mothers and their infants.

## 1. Introduction

Perinatal depression is defined as a non-psychotic depressive episode occurring during pregnancy (antenatal depression) or following childbirth (postpartum depression), up to one year postnatally [1]. The most common symptoms of perinatal depression are distinct from the commonly experienced “baby blues”, which usually end within the first two weeks after birth [2]. 

Perinatal depression represents a high worldwide burden [3]. The majority of evidence comes from high-income countries, where the prevalence has recently been estimated at 11.4% [4]. In contrast, in the low- and middle-income countries (LMICs) there is a lack of research and therefore few global estimates. Prevalence in these countries is in the range of 10–41% depending on the context, the perinatal stage and the diagnostic instruments used [5]. However, reported data are only available from a minority of these countries (<10%) [6]. 

Various hypotheses have been advanced to explain the high prevalence of perinatal mental disorders in LMICs [7]. Sociodemographic factors such as age and marital status, financial problems, lack of social support and history of mental health problems appear to be linked with poor mental health [6]. This high prevalence is also explained by the high rate of adolescent pregnancy in sub-Saharan African countries, reaching 19.3% [8]. This high prevalence also represents a significant risk factor for psychiatric disorders [9], which are likely to negatively affect the lives of both the mother and the child. Evidence suggests that maternal depression has negative impacts on the infant’s weight gain and stunting, cognitive development and overall health concerns [10]. 

Despite its known adverse consequences, perinatal depression is still underdiagnosed and undertreated in LMICs [11]. There is also a lack of prevention and treatment programs for perinatal depression [12]. To address this problem, it is necessary to develop interventions to screen mothers in the antenatal and postnatal periods [13]. Perinatal depression should be recognized as a public health issue and be included in health policies for women and infants. Addressing psychological distress during the perinatal period is one way to reduce the risks for the mother and her child. Adapted maternal care programs should be developed according to the identified needs of the community [9].

Cameroon is a lower-middle-income country in which almost one-quarter of adolescent women aged 15 to 19 are already mothers or are pregnant with their first child [14]. Despite this high prevalence of teenage mothers, there is a lack of data on perinatal depression among these women, and on its impact on the development of the newborn. In order to develop effective and targeted health programs to support pregnant teenagers and young mothers, we carried out a psychosocial diagnostic assessment on a sample drawn from this population. The aim of this cross-sectional study was to provide baseline data on the sociodemographic characteristics of this population and their influence on the occurrence of depressive symptoms. Risk factors were assessed, and the prevalence of depressive disorders was determined.

## 2. Methods

### 2.1. Setting

This study is part of a program introducing psychosocial care for teenage mothers in collaboration with the Ministry of Public Health (MINSANTE), Ministry for the Promotion of Woman and Family (MINPROFF), a local community association called RENATA (Reseau National des Tantines) promoting prevention of teenage pregnancies [15] and UniPsy (Uni-Psy et Bien-être), an association specializing in mental health care. The main mission of the RENATA association is to educate young women about sexual and reproductive health and to raise their awareness of sexual violence issues. The association also defends the interests of teenage mothers [16]. The psychosocial care program in favor of teenage mothers was introduced by the Institute of Global Health of the university of Geneva in collaboration with ASP, a Non-Governmental Organization (NGO) in Geneva. First, a group of clinical psychologists received specialist training in mother and child mental health issues and care. As a next step, first-line health workers including health workers from MINSANTE, social workers from MINPROFF, social collaborators from RENATA, psychologists and psychology students from UniPsy followed a training program on mental healthcare for primary health care providers, developed by WHO (the World Health Organization) called mhGAP (Mental health Global Action Program).

### 2.2. Population

Participants were recruited by the first-line health and community workers, the supervising psychologists and social collaborators from RENATA between 2014 and 2017 in three different contexts: through door-to-door invitations; health facilities, associations and groups; and via the health workers’ personal networks. The study population included 1344 women aged 20 years or less in the perinatal period (from early pregnancy until the end of the child’s first year of life). We excluded data from participants who were older than 20 years, those who did not fully complete the EPDS scale or any of the additional questions, and from participants with children aged 13–24 months. Only participants who had given their written consent to participate were included in the study.

A total of 1307 women fulfilled the inclusion criteria and were therefore included in the study and data analysis.

### 2.3. Study Design

The study was conducted by the Geneva-based NGO ASP (Actions en Santé Publique) collaborating with a newly created group of psychotherapists called UniPsy in different areas of Yaoundé, Cameroon. The data were collected by the project team, who were trained for mental health in primary health care by ASP and UniPsy. When they met pregnant teenagers, the team members invited the young women to answer a questionnaire to assess their risk of perinatal depression. Participants were informed that ASP/UniPsy could provide a free psychiatric care program in case of high risk. 

### 2.4. Study Tool

Data were collected using mixed interviews (both directive and semi-directive) with four sections, at two time points. The first part included participants’ sociodemographic characteristics: age, marital status, religion, perinatal status, stage of pregnancy and age of their child. The second part concerned a psychosocial and mental health assessment. The third part consisted of the Edinburgh Postnatal Depression Scale (EPDS) questionnaire, which was used as a perinatal depression screening tool. The fourth part focused on perinatal depression risk factors.

### 2.5. Edinburgh Postnatal Depression Scale (EPDS)

The Edinburgh Postnatal Depression Scale (EPDS) is the most widely used screening tool for perinatal depression [17]. It was originally developed in 1987 by Cox et al. to detect postnatal depression, but it is also recommended for use during pregnancy [18,19]. The EPDS is a 10-item self-report questionnaire in which each question is scored from 0 to 3 (resulting range, 0–30). According to the supporting literature, a cut-off of 12/13 is taken to indicate “possible depression” [20]. A cut-off of 12 was chosen in this study for both post- and antenatal period to indicate the probability of depressive illness [17]. A study by Bianciardi indicates that this choice of cut off can be used in a context such as ours, during the pregnancy and postpartum period [21]. This instrument is not a diagnostic tool. A clinical assessment is required to make a definitive diagnosis [20]. The only EPDS version used in this study was the translated and validated French version [22].

### 2.6. Clinical Assessment

The assessment was done during two interviews. We assessed the perinatal history (history and conditions of pregnancy/birth), parenthood (representations of the mother about the baby, mother–baby relationship, the mother’s own childhood and relationships with parents), signs and symptoms of depression, risk factors of perinatal depression and social needs. The EPDS questionnaire was also introduced. All these items were discussed jointly with the supervising clinical psychologist. The diagnosis of depression was systematically carried out according to the mhGAP guide and based on DSM 5 criteria (Diagnostic and Statistical Manual of Mental Disorders). 

### 2.7. Sample Size

This study is based on a pragmatic sample. Subjects were recruited from accessible places by the project team. In accordance with our exclusion criteria, a total of 1307 women took part in the study.

### 2.8. Statistical Analyses

Statistical analyses were run using STATA 14 (Stata V14, StataCorp., College Station, TX, USA). The normality of distribution was tested by the Kolmogorov–Smirnov test. Descriptive statistics and frequencies were analyzed for all variables with 95% confidence intervals. 

Multivariate logistic regression was performed to identify factors associated with perinatal depression status and sociodemographic factors. The status of possible perinatal depression (EPDS score <12 or ≥12) was used as the primary outcome. In multivariate models, only the covariates that were of a priori interest were included in a univariate analysis. A *p*-value of less than 0.05 was accepted as statistically significant. 

### 2.9. Ethical Approval

The study protocol was approved by the National Ethics Committee on Research for Human Health (NECRHH) in Yaoundé, approval number: 2014/03/436/L/CNERSH/SP. All participants signed an informed consent form prior to participating in the study, and were informed that they could withdraw from the study at any time.

## 3. Results

### 3.1. Sociodemographic Characteristics of Participants

An initial sample of 1344 women agreed to participate in the study; of these, data were incomplete for 37 women. Thus, the study sample was a total of 1307 women. Participants’ baseline characteristics are presented in Table 1. 

The participants’ mean age was 17.4 (range 13–20). At the time of the study, most women were single (78%). Other participants were cohabitating (17%) or married (5%). Most of the women were Christians (68.9% Catholics, 16.9% Protestants and 5.7% Pentecostalists), and only 4.9% were Muslims or of another religion (3.6%). At the time of assessment, 48.4% of the women were pregnant and 51.6% were in the postpartum period. Among the pregnant women, 18% were in the first trimester of pregnancy, 45.1% in the second and 36.9% in the third. The majority of women who had given birth were in the first year after childbirth, 37.9% were in the early post-partum period with a child under 3 months old, 26.6% had a child aged between 4 and 6 months and 35.5% had a child between 7 and 12 months of age. 

### 3.2. Perinatal Depression Risk Factors

The full description of the EDPS scores and the results of the different dimensions of this scale are presented in Table 2.

Results of the logistic regression predicting perinatal depression risk factors are presented in Table 3. The prevalence of depressive disorder symptoms in our sample of teenage mothers or young pregnant women in Yaoundé is 70.0%. Unintended or unplanned pregnancy significantly increased the risk of having a high EPDS score (aOR: 1.33, 1.14–1.56 CI95%). The risk was also significantly increased for single or separated women (aOR: 1.34, 1.12–1.60 CI95%). Women who suffered from depression and anxiety before childbirth were more likely to have a high EPDS score (aOR: 1.50, 1.02–2.27 CI95%), as were women who had experienced an abortion (aOR: 2.60, 1.03–7.14 CI95%). Domestic violence was also associated with a high EPDS score (aOR: 1.76, 1.12–2.83 CI95%). There were no statistically significant differences between the EPDS score with regard to social risk factors.

## 4. Discussion

The results of this study suggest a high prevalence of depressive disorders in a sample of teenage mothers and young pregnant women from Yaoundé, Cameroon. Among 1307 teenage women 915 (70.0%) were found to have a score above or equal to 12. Women who scored above this threshold are more likely to be suffering from depressive disorders of varying severity [17]. According to international scientific literature, teenage mothers are twice as likely to suffer from postpartum depression than their adult counterparts [23]. In LMICs there is a lack of data estimating the prevalence of perinatal depression despite the high burden of mental health problems during the perinatal period, especially in adolescents [6]. A recent study conducted in Kenya reported a prevalence of 32.5% for adolescents reporting clinically elevated depression symptoms using the EPDS [24], which is much lower than the prevalence of 70% found in the present study. Another project conducted in Cameroon (also using the EPDS) reported a prevalence of postpartum depression of 23.4%. However, the prevalence may differ from the current study since the majority of women were over 20 years of age.

The present study identified five significant risk factors for a high EPDS score, including being single or separated, unplanned pregnancy, depression or severe anxiety before childbirth, history of abortion and domestic violence.

Being single or separated significantly increased the risk of having a high EPDS score (aOR: 1.34, 1.12–1.60 CI95%) compared to other teenagers who were living in cohabitation or married. Absence of support from a partner is one of the risk factors often cited in studies [25] which increases the risk of depressive disorders. Moreover, the stigma associated with being single may contribute to poorer mental health. A supportive husband and a stable marital relationship play an important role in promoting a strong social support system for the mother [25]. The high percentage of single or separated women included in this study (78.0%) could explain the large number of participants with high EPDS scores.

Perinatal depression was associated with pregnancy-related circumstances such as unplanned pregnancy. Unintended pregnancy does not necessarily mean an unwelcome childbirth, but it certainly affects the life of a young woman. In the present sample, the teenage mothers were 1.33 times more likely to have a high EPDS score if they had experienced an unplanned pregnancy. This result is consistent with findings from previous studies showing that unplanned pregnancies are twice as likely to be associated with psychiatric disorders such as depression, anxiety and perinatal depression [26]. 

Whilst depression or severe anxiety before childbirth significantly affects the risk of subsequent depression, there is very little research on antenatal mental health and associated risk factors in African women. This is an important gap in the literature that must be addressed, particularly as prenatal psychological health has been identified as a strong predictor for postnatal mental health [27]. However, this finding is not surprising because anxiety and depression are generally directly related to other psycho-social and environmental factors. In this study the other significant risk factors identified are also strongly associated with mental health problems [28]. Although a lack of social support has been linked with a higher prevalence of perinatal depression in other studies [29,30], no significant results were found in this study. 

We observed a significant increase in the risk of depressive disorders in women with a history of abortion. This risk was increased by 2.6 times when compared to women with a history of miscarriage or stillbirth. These results should be interpreted with caution because they concern a very small percentage of women included in this study. However, the findings appear consistent with results from previous studies which have demonstrated an increased risk of depression and anxiety, including post-traumatic stress disorder, after childbirth [31,32]. This association remains constant across the pre- and postnatal periods, which means that depressive disorders and anxiety persist after childbirth, even if the child is healthy [31]. 

Low-income women have a greater risk of miscarriage and often experience more than one loss/abortion. In Cameroon, the prevalence of voluntary induced abortion remains high [33], which can significantly increase the risk of depressive disorders in this population. Further research is necessary to understand why previous miscarriage or abortion is associated with perinatal depression among the population of teenage women. 

Among the risk factors related to family and marital relationship, domestic violence is the only factor that significantly increases the likelihood of depressive disorders. Domestic violence is a common problem in society, especially in Cameroon where its prevalence has recently been estimated at 40% [34]. Additionally, women under the age of 20 are reported to be at even higher risk of experiencing domestic violence during pregnancy than older women [35]. One of the main consequences of domestic violence is a negative impact on mental health, particularly depressive disorders [36]. This is confirmed in the current study, the results of which show a 1.8-fold increase in the risk of depressive disorders during the perinatal period among the sample population. This finding is consistent with previous studies conducted in LMICs which demonstrate an increased risk of post-partum depression in victims of domestic violence [37]. No statistically significant results were found concerning polygamy, which remains a fairly common practice in Cameroon, or difficulties with in-laws. 

### Strengths and Limitations of the Study

The strengths of this study include the large study population which included 1307 teenage mothers or pregnant women from Yaoundé, Cameroon. The recruitment method made it possible to include young women from all backgrounds. The door-to-door recruitment helped to include women who would not have attended a community health center. 

This study also has several limitations. Although a validated French version of the EPDS is available, this tool has not been validated for use in developing countries. In fact, there is a lack of data concerning the local cultural sensitivity of this questionnaire in LMICs [38]. Misunderstanding over intended meanings and comprehension of the questions seem to be the main barriers to its use in countries like Cameroon [39]. The tool was originally developed to assess postpartum depression. The test validity for the antenatal period still needs to be improved. Furthermore, it does not include a somatic subscale, which can present a further barrier to use in developing countries where psychological disorders can be considered as somatic symptoms [40]. 

## 5. Conclusions

In this study, the prevalence of perinatal depression among teenagers and young mothers was found to be high when compared to existing data. Nonetheless, the risk factors were comparable to those for women aged over 20 years. The prevalence of perinatal depression in this young female population was higher than that for women in developed countries, but the risk factors are broadly comparable. Further studies are needed to confirm these results—particularly longitudinal studies following women over the course of many months, which would make it possible to compare the risk of developing mental health problems before and after childbirth. 

## Figures and Tables

**Table 1 jcm-10-04164-t001:** Sociodemographic characteristics of the study population.

	*N*	%	CI95%
Total			
Age (*n* = 1307)			
Mean 17.4			
13–15 years	62	4.7%	3.9–6.3
16–17 years	323	24.7%	22.6–27.3
18–20 years	922	70.5%	67.4–72.3
Marital status (*n* = 1307)			
Single	1017	77.8%	75.6–80.1
Cohabiting	220	16.8%	14.7–18.7
Married	70	5.4%	4.1–6.5
Religion (*n* = 1307)			
Catholic	900	68.9%	65.8–70.9
Protestant	221	16.9%	15.1–19.1
Pentecostalist	75	5.7%	4.6–7.2
Muslim	64	4.9%	3.7–6.1
Other	47	3.6%	2.8–4.9
Perinatal status (*n* = 1307)			
Pregnant	632	48.4%	45.7–51.1
Post-partum	675	51.6%	48.8–54.2
Stage of pregnancy (*n* = 632)			
First trimester	114	18.0%	14.1–20.1
Second trimester	285	45.1%	41.2–49.1
Third trimester	233	36.9%	34.1–41.8
Age of the child (=657)			
≤3 months	249	37.9%	33.2–40.6
4–6 months	175	26.6%	23.2–30.1
7–12 months	233	35.5%	30.7–38.1

**Table 2 jcm-10-04164-t002:** Description of EDPS scores and categories.

	*N*	%	CI95%	
I could laugh and look on the bright side				
As much as possible	488	37.22	34.64	39.87
Not quite so often as usual	361	27.54	25.17	30.00
A lot less often than usual	313	23.87	21.63	26.24
Not at all	149	11.37	9.73	13.17
I felt confident when thinking about the future				
As much as usual	498	38.04	35.44	40.70
Somewhat less than usual	387	29.56	27.14	32.12
A lot less than usual	263	20.09	17.99	22.33
Not at all	161	12.30	10.60	14.16
I blamed myself without any reason				
Yes, most of the time	353	27.03	24.67	29.49
Yes, sometimes	560	42.88	40.21	45.58
Not very often	256	19.60	17.52	21.82
No, not at all	137	10.49	8.91	12.24
I felt anxious/worried for no reason				
No, not at all	265	20.38	18.26	22.64
Almost never	165	12.69	10.96	14.59
Yes, sometimes	624	48.00	45.29	50.72
Yes, very often	246	18.92	16.86	21.12
I felt scared/panicked for no reason				
Yes, very often	200	15.30	13.43	17.33
Yes, sometimes	510	39.02	36.40	41.69
Not very often	296	22.65	20.44	24.98
No, not at all	301	23.03	20.81	25.37
I felt overwhelmed by events				
Yes, most of the time I felt unable to cope with situations	512	39.11	36.50	41.78
Yes, sometimes I didn’t feel able to cope with situations	350	26.74	24.39	29.19
No, I could cope with most situations	305	23.30	21.07	25.65
No, I felt as efficient as usual	142	10.85	9.25	12.62
I felt so unhappy that it caused me sleep problems				
Yes, most of the time	243	18.65	16.60	20.83
Yes, sometimes	451	34.61	32.06	37.23
Not very often	288	22.10	19.91	24.42
No, not at all	321	24.64	22.35	27.03
I felt sad/unhappy				
Yes, most of the time	314	24.02	21.77	26.40
Yes, sometimes	449	34.35	31.81	36.96
Not very often	333	25.48	23.17	27.89
No, not at all	211	16.14	14.22	18.21
I felt so unhappy that I cried				
Yes, most of the time	306	23.45	21.21	25.81
Yes, very often	336	25.75	23.43	28.17
Only from time to time	381	29.20	26.78	31.71
No, never	282	21.61	19.44	23.90
There are times that I thought about hurting myself				
Yes, very often	112	8.61	7.18	10.23
Sometimes	304	23.37	21.13	25.72
Almost never	171	13.14	11.39	15.06
Never	714	54.88	52.17	57.57
What is your EPDS score?				
0 to 8	242	18.52	16.48	20.69
9 to 11	150	11.48	9.83	13.29
12 to 30	915	70.01	67.48	72.45

**Table 3 jcm-10-04164-t003:** Perinatal depression risk factors.

	EPDS Score < 12 (*n* = 392)	EPDS Score ≥ 12 (*n* = 915)	OR (CI95%)	aOR (CI95%)
	*N* (%)	*N* (%)		
Circumstances of pregnancy/maternity *				
Teenage pregnancy	379 (96.7)	900 (98.4)	Referent	
Unintended/unplanned pregnancy	292 (74.5)	802 (87.7)	1.16 (0.97–1.39)	1.33 (1.14–1.56)
Single or separated	189 (48.2)	524 (57.3)	1.17 (0.95–1.43)	1.34 (1.12–1.60)
Number of children > 3	34 (8.7)	63 (6.9)	0.78 (0.51–1.20)	0.95 (0.68–1.37)
Birth of a girl in cultures preferring a boy	5 (1.3)	21 (2.3)	1.77 (0.66–4.73)	1.94 (0.83–4.90)
Health issues *				
History of mental illness	33 (8.4)	130 (14.2)	Referent	
Depression or severe anxiety before childbirth	67 (17.1)	346 (37.8)	1.31 (0.83–2.08)	1.50 (1.02–2.27)
Illness during pregnancy or childbirth	92 (23.5)	296 (32.3)	0.82 (0.52–1.28)	1.01 (0.71–1.47)
Social risk factors *				
Lack of practical support	73 (18.6)	443 (48.4)	Referent	
Poverty and lack of financial resources	145 (37)	591 (64.6)	0.67 (0.49–0.91)	0.79 (0.61–1.03)
Refugee/migrant	7 (1.8)	16 (1.7)	0.38 (0.15–0.95)	0.50 (0.27–1.07)
War zone, conflict or natural disaster	5 (1.3)	10 (1.1)	0.33 (0.11–0.99)	0.45 (0.23–1.11)
Negative childbirth/procreation experience *				
History of miscarriage or stillbirth	18 (4.6)	66 (7.2)	Referent	
Seriously ill baby, deformed baby	6 (1.5)	24 (2.6)	1.09 (0.39–3.07)	1.29 (0.59–3.27)
Abortion	5 (1.3)	44 (4.8)	2.40 (0.83–6.94)	2.60 (1.03–7.14)
Family and marital relationship *				
Unsupportive/absent husband	84 (21.4)	414 (45.2)	Referent	
Polygamous family	23 (5.9)	72 (7.9)	0.64 (0.38–1.07)	0.77 (0.51–1.20)
Domestic violence	22 (5.6)	177 (19.3)	1.63 (0.99–2.70)	1.76 (1.12–2.83)
Inability to confide in partner	75 (19.1)	316 (34.5)	0.85 (0.61–1.21)	0.98 (0.74–1.34)
Difficult in-law relationships	52 (13.3)	244 (26.7)	0.95 (0.65–1.39)	1.08 (0.78–1.52)

* Many possible answers. Statistically significant results are in bold. Odds ratios were adjusted for the following co-variable: age, marital status, religion, perinatal status, stage of pregnancy and age of child.

## Data Availability

The data presented in this study are available on request from the 290 corresponding author. The data are not publicly available due to privacy reasons.

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
