# Peer review of "Teenage Mothers in Yaoundé, Cameroon—Risk Factors and Prevalence of Perinatal Depression Symptoms"

_jcm, 2021, doi:10.3390/jcm10184164_

Round 1
Reviewer 1 Report
Thank you for an opportunity to review your valuable research. This study shows that which factors are associated with perinatal depressive symptoms among teenage pregnancy in Cameroon. To improve maternal mental health, it is very important to find which factors are risky or preventable, especially, teenage mothers are vulnerable so they should be supported.
This study collected very valuable population, it was used adequate postpartum depressive symptoms’ index, and it had important message for mental health of teenage mothers. However, there are significant errors in the methods and results section. Major or minor comments are detailed below.
In method section (page 3, line 97-103), there are less explanation of gathering participants. In data collection, please add up the information of how many people were collected, excluded, and were used in this study. How did the authors treat the missing data? The authors should provide this information.
I also wonder the language of questionnaire. Did the authors use questionnaires only French or both French and English, or other languages? If you translated special languages, how did you treat or confirm the translation bias? The authors should be explained of this information.
In method section(page 3, line 111-113), the authors give to readers IRB information in ‘Ethical approval’ section(page 4, line 156-160) so I think it is not necessary to explain here. Please note the IRB approval numbers in the ‘Ethical approval’ section.
In the statistical analysis, and Table 1, what is IC 95%? Does it mean 95% CI? I think it feels awkward explanation. Please revise or clarify it.
In table 1, please show the distribution of all variables, and please add the distribution of EDPS scores.
Please add the specific information of covariates. The authors mentioned in the multiple logistic model adjusted for covariates but there was no information of these. Also, please add the foot notes in the table 2 which covariates adjusted.
In table 2, I wonder the author treat this model. In the circumstance of pregnancy/maternity, does the authors analyzed each variable combined? Why teenager pregnancy is reference group? The authors interpret unintended pregnancy is higher risk of postpartum depression compared to teenage pregnancy? I am so confused this result. If the authors think so, it was wrong analysis. In addition, it doesn’t make sense which factors adjusted. Please re-analysis all multiple model adjusted for all covariates.
In discussion section (page 7, page 213-214), single or separated teen mothers are high risk of EPDS score compared to those who? It is important and basic sentence when you use ‘ratio’.
Author Response
Comment 1 : Thank you for an opportunity to review your valuable research. This study shows that which factors are associated with perinatal depressive symptoms among teenage pregnancy in Cameroon. To improve maternal mental health, it is very important to find which factors are risky or preventable, especially, teenage mothers are vulnerable so they should be supported.
This study collected very valuable population, it was used adequate postpartum depressive symptoms’ index, and it had important message for mental health of teenage mothers. However, there are significant errors in the methods and results section. Major or minor comments are detailed below.
Answer 1 : Thank you very much for your interest in this article and for the opportunity to improve it by responding to your comments.
Comment 2 : « In method section (page 3, line 97-103), there are less explanation of gathering participants. In data collection, please add up the information of how many people were collected, excluded, and were used in this study. How did the authors treat the missing data? The authors should provide this information.
»
Answer 2 : We have added a paragraph in the method section to explain recruitment more clearly.
Comment 3 : I also wonder the language of questionnaire. Did the authors use questionnaires only French or both French and English, or other languages? If you translated special languages, how did you treat or confirm the translation bias? The authors should be explained of this information.
»
Answer 3: We have added a paragraph in the method section to provide more i detail aboutthe version and language of EDPS.
Comment 3 : In method section(page 3, line 111-113), the authors give to readers IRB information in ‘Ethical approval’ section(page 4, line 156-160) so I think it is not necessary to explain here. Please note the IRB approval numbers in the ‘Ethical approval’ section.
»
Answer 3: We have modified this part and added the IRB approval number.
Comment 3 : In the statistical analysis, and Table 1, what is IC 95%? Does it mean 95% CI? I think it feels awkward explanation. Please revise or clarify it.
»
Answer 3: We apologise for the error here and we have now amended table 1.
Comment 4 : In table 1, please show the distribution of all variables, and please add the distribution of EDPS scores.
Answer 4: We have added a table which gives a description of the variables not listed in
Table 1 and this includes a description of the EDPS score
Comment 5 : Please add the specific information of covariates. The authors mentioned in the multiple logistic model adjusted for covariates but there was no information of these. Also, please add the foot notes in the table 2 which covariates adjusted.
Answer 5: we have added the information requested in Table 3
Comment 6 : In table 2, I wonder the author treat this model. In the circumstance of pregnancy/maternity, does the authors analyzed each variable combined? Why teenager pregnancy is reference group? The authors interpret unintended pregnancy is higher risk of postpartum depression compared to teenage pregnancy? I am so confused this result. If the authors think so, it was wrong analysis. In addition, it doesn’t make sense which factors adjusted. Please re-analysis all multiple model adjusted for all covariates.
Answer 7 : For the multivariate analysis we followed the advice of the research psychologist of the Renata team who advised us on the statistical analysis. For the Pregnancy/Maternity case, no we did not analyse the two groups separately from each other. The teenage pregnancy category was chosen as the reference group arbitrarily because it was the category with the largest N, and on the advice of the statistician.
Comment 8: In discussion section (page 7, page 213-214), single or separated teen mothers are high risk of EPDS score compared to those who? It is important and basic sentence when you use ‘ratio’.
Answer 8: We have ammended this sentence so that it is clearer.
Reviewer 2 Report
Teenage mothers in Yaoundé, Cameroon – Risk Factors and 2 Prevalence of Perinatal Depression symptoms
I much appreciated this paper but I suggested some revisions.
- Please replace DSM V with DSM 5 (line 26, 139..)
- English needs a revision, in particular the discussion and conclusion are hard to follow.
- I suggest authors to exclude the 14 participants with children 13-24 months. According to DSM 5 criteria, perinatal depression may emerge during pregnancy and 4 weeks after partum Even though it is well-known that this disorder may appear during the first 12 months after partum. For these reason, it is appropriate to remove the 14 women from your sample.
- In table 2: specify “large number of children”; chenge large with >3….
- The EPDS score is the main concern and limitation of your study. EPDS may be used with different cut-off during pregnancy and postpartum as I wrote in a recent paper, you may referred to this manuscript with the aim of justify the cut-off used. (The anxious aspects of insecure attachment styles are associated with depression either in pregnancy or in the postpartum period) DOI: 10.1186/s12991-020-00301-7
Accordingly, remove lines 277-278 and 280-281. I do not think this is a limitation because you used a validated cut-off and questionnaire. You only need to substantiate your choice using literature.
- Line133: We assessed the perinatal history
- Line 203: ..in Kenya found reported
- Line 205: this prevalence is lower copared to the prevalence of 70% that we found in this study
- Line 209: here you can discuss about EPDS cut-off
- Line 219: is an element
- Line 221: perinatal depression was predominantly also associated
- Line 227: replace: we found that previous psychiatric disorders such as depression and anxiety increased the risk of perinatal depression
- Line 231-233: it is not clear, explain
- Line 253-260: i suggest you to argument that trauma (IPV) increased the risk of depression via biological pathway, affecting markers of inflammation (doi: 10.1186/s12991-020-00322-2).
- Line 277-278: delete the sentence “the test validity…” (you will discuss it after line 209)
- Line 280-281: delete it
- Line 283: in this study the prevalence of perinatal depression among adolescent mothers was higher compared to previous data. Nonetheless, risk factors were comparable to non-adolescent women.
- Line 284-289: delete it
- Explain the implication of your results: having a clear picture of this alarming prevalence it is crucial improving the screening of perinatal depression in adolescent women and preventing risk factors. This is particularly compelling considering the well-known disabling mid and long-term consequences of PeriNatal Depression for both mother and offspring …
Author Response
Comment 1: « Teenage mothers in Yaoundé, Cameroon – Risk Factors and 2 Prevalence of Perinatal Depression symptoms. I much appreciated this paper but I suggested some revisions.»
Answer 1 : Thank you very much for your positive feedback and helpful suggestions. . We have done our best toupdate the paper, in keeping with your comments..
Comment 2: « Please replace DSM V with DSM 5 (line 26, 139..).»
Answer 2 : We have made this modification.
Comment 3: « English needs a revision, in particular the discussion and conclusion are hard to follow.
Answer 3 : The whole article has now been proofread by an English-speaking editor and the grammar has been revised, where necessary..
Comment 4: « I suggest authors to exclude the 14 participants with children 13-24 months. According to DSM 5 criteria, perinatal depression may emerge during pregnancy and 4 weeks after partum Even though it is well-known that this disorder may appear during the first 12 months after partum. For these reason, it is appropriate to remove the 14 women from your sample..
Answer 4 : Thank you for this relevant comment. As per your suggestion, we have excluded these 14 participants and modified Table 1.
Comment 5: In table 2: specify “large number of children”; chenge large with >3….
.
Answer 5 : We have made this modification.
Comment 6: The EPDS score is the main concern and limitation of your study. EPDS may be used with different cut-off during pregnancy and postpartum as I wrote in a recent paper, you may referred to this manuscript with the aim of justify the cut-off used. (The anxious aspects of insecure attachment styles are associated with depression either in pregnancy or in the postpartum period) DOI: 10.1186/s12991-020-00301-7
Accordingly, remove lines 277-278 and 280-281. I do not think this is a limitation because you used a validated cut-off and questionnaire. You only need to substantiate your choice using literature.
Answer 6: We have modified this paragraph, added this reference and deleted the requested lines.
Comment 7:
Line133: We assessed the perinatal history
Line 203: ..in Kenya found reported
Line 205: this prevalence is lower copared to the prevalence of 70% that we found in this study
Line 209: here you can discuss about EPDS cut-off
Line 219: is an element
Line 221: perinatal depression was predominantly also associated
Line 227: replace: we found that previous psychiatric disorders such as depression and anxiety increased the risk of perinatal depression
Line 231-233: it is not clear, explain
Line 253-260: i suggest you to argument that trauma (IPV) increased the risk of depression via biological pathway, affecting markers of inflammation (doi: 10.1186/s12991-020-00322-2).
Line 277-278: delete the sentence “the test validity…” (you will discuss it after line 209)
Line 280-281: delete it
Line 283: in this study the prevalence of perinatal depression among adolescent mothers was higher compared to previous data. Nonetheless, risk factors were comparable to non-adolescent women.
Line 284-289: delete it
.
Answer 6 : We have made the above modifications.
Reviewer 3 Report
Dear Authors,
Your study covers a problem which is especially important in Cameroon in which as you stated "almost one-quarter of adolescent women age 15 to 19 are already mothers or are pregnant with their first child". The paper is interesting however I have some comments which should be explained:
- First, the number of participants is confusing. You stated that 1,344 women answered part of a questionnaire which included questions on their socio-demographic background, however the EPDS score could be evaluated on a total of 1,307 women. On the basis of the EPDS assessment you concluded that 70% of women had depressive disorder symptoms. My question is: Did remaining 37 women (1344-1307) have depression? In my opinion, only the 1,307 women interviewed by the EPDS should be analyzed. This could solve the problem of different n numbers for Age (n = 1333), Marital status (n = 1329), Religion and mainstreams (n = 1318) and Perinatal status (n = 1316) [Table 1].
- Table 2 is too long. It should be reformatted. The two middle N and % columns can be combined into one - N(%).
- In the notation of the number of women - 1'344 (Abstract and Introduction) there should be no apostrophe but a comma. Be consistent in writing other numbers.
- Lines 164, 183, 186, 188 and 220 - you used EDPS acronym while it should be EPDS.
Author Response
Comment 1: « Your study covers a problem which is especially important in Cameroon in which as you stated "almost one-quarter of adolescent women age 15 to 19 are already mothers or are pregnant with their first child". The paper is interesting however I have some comments which should be explained:.»
Answer 1 : Thank you very much for your positive feedback and helpful suggestions. . We have done our best toupdate the paper, in keeping with your comments.
Comment 2: « First, the number of participants is confusing. You stated that 1,344 women answered part of a questionnaire which included questions on their socio-demographic background, however the EPDS score could be evaluated on a total of 1,307 women. On the basis of the EPDS assessment you concluded that 70% of women had depressive disorder symptoms. My question is: Did remaining 37 women (1344-1307) have depression? In my opinion, only the 1,307 women interviewed by the EPDS should be analyzed. This could solve the problem of different n numbers for Age (n = 1333), Marital status (n = 1329), Religion and mainstreams (n = 1318) and Perinatal status (n = 1316) [Table 1].
Answer 2 : We fully agree that these different Ns in the results section and in Table 1 were confusingWe only retained a sample of 1,307 women who had answered the EDPS questionnaire. We have therefore modified the method and results section to include only this sample.
Comment 3: Table 2 is too long. It should be reformatted. The two middle N and % columns can be combined into one - N(%).
Answer 3 : We have modified and simplified Table 2 as requested by the referee.
Comment 4: In the notation of the number of women - 1'344 (Abstract and Introduction) there should be no apostrophe but a comma. Be consistent in writing other numbers.
Answer 4 : We have modified these notations.
Comment 5: Lines 164, 183, 186, 188 and 220 - you used EDPS acronym while it should be EPDS.
Answer 5 : We have modified these acronyms.
Round 2
Reviewer 1 Report
The authors sufficiently improved their manuscript and adequately responded the reviewers' comments.
This manuscript is a resubmission of an earlier submission. The following is a list of the peer review reports and author responses from that submission.